# Untangling in Invariant Speech Recognition

**Cory Stephenson**
Intel AI Lab
cory.stephenson@intel.com

**Jenelle Feather**
MIT
jfeather@mit.edu

**Suchismita Padhy**
Intel AI Lab
suchismita.padhy@intel.com

**Oguz Elibol**
Intel AI Lab
oguz.h.elibol@intel.com

**Hanlin Tang**
Intel AI Lab
hanlin.tang@intel.com

**Josh McDermott**
MIT/ Center for Brains, Minds, and Machines
jhm@mit.edu

**SueYeon Chung**
Columbia University/ MIT
sueyeon@mit.edu

## Abstract

Encouraged by the success of deep neural networks on a variety of visual tasks, much theoretical and experimental work has been aimed at understanding and interpreting how vision networks operate. Meanwhile, deep neural networks have also achieved impressive performance in audio processing applications, both as sub-components of larger systems and as complete end-to-end systems by themselves. Despite their empirical successes, comparatively little is understood about how these audio models accomplish these tasks. In this work, we employ a recently developed statistical mechanical theory that connects geometric properties of network representations and the separability of classes to probe how information is untangled within neural networks trained to recognize speech. We observe that speaker-specific nuisance variations are discarded by the network's hierarchy, whereas task-relevant properties such as words and phonemes are untangled in later layers. Higher level concepts such as parts-of-speech and context dependence also emerge in the later layers of the network. Finally, we find that the deep representations carry out significant temporal untangling by efficiently extracting task-relevant features at each time step of the computation. Taken together, these findings shed light on how deep auditory models process time dependent input signals to achieve invariant speech recognition, and show how different concepts emerge through the layers of the network.

## 1 Introduction

Understanding invariant object recognition is one of the key challenges in cognitive neuroscience and artificial intelligence[1]. An accurate recognition system will predict the same class regardless of stimulus variations, such as the changes in viewing angle of an object or the differences in pronunciations of a spoken word. Although the class predicted by such a system is unchanged, the internal representations of individual objects within the class may differ. The set of representations corresponding to the same object class can then be thought of as an object manifold. In vision systems, it has been hypothesized that these "object manifolds", which are hopelessly entangled in the input, become "untangled" across the visual hierarchy, enabling the separation of different categories both in the brain [2] and in deep artificial neural networks [3]. Auditory recognition also requires the separation of highly variable inputs according to class, and could involve the untangling of 'auditory

class manifolds'. In contrast to vision, auditory signals unfold over time, and the impact of this difference on underlying representations is poorly understood.

Speech recognition is a natural domain for analyzing auditory class manifolds not only with word and speaker classes, but also at the phonetic and semantic level. In recent years, hierarchical neural network models have achieved state of the art performance in automatic speech recognition (ASR) and speaker identification [4, 5]. Understanding how these end-to-end models represent language and speech information remains a major challenge and is an active area of research [6, 7]. Several studies on speech recognition systems have analyzed how phonetic information is encoded in acoustic models [8, 9, 10], and how it is embedded across layers by making use of classifiers [11, 12, 13, 6]. It has also been shown that deep neural networks trained on tasks such as speech and music recognition resemble human behavior and auditory cortex activity [14]. Ultimately, understanding speech-processing in deep networks may shed light on understanding how the brain processes auditory information.

Much of the prior work on characterizing how information is represented and processed in deep networks and the brain have focused on linear separability, representational similarity, and geometric measures. For instance, the representations of different objects in vision models and the macaque ventral stream become more linearly separable at deeper stages, as measured by applying a linear classifier to each intermediate layer [2, 3]. Representations have been compared across different networks, layers, and training epochs using Canonical Correlation Analysis (CCA) [15, 16, 17]. Representational similarity analysis (RSA), which evaluates the similarity of representations derived for different inputs, has been used to compare networks [18, 19]. Others have used explicit geometric measures to understand deep networks, such as curvature [3, 20], geodesics [21], and Gaussian mean width [22]. However, none of these measures make a concrete connection between the separability of object representations and their geometrical properties.

In this work, we make use of a recently developed theoretical framework[23, 24, 25] based on the replica method [26, 27, 28] that links the geometry of object manifolds to the capacity of a linear classifier as a measure of the amount of information stored about object categories per feature dimension. This method has been used in visual convolutional neural networks (CNNs) to characterize object-related information content across layers, and to relate it to the emergent representational geometry to understand how object manifolds are untangled across layers [25]. Here we apply manifold analyses[1] to auditory models for the first time, and show that neural network speech recognition systems also untangle speech objects relevant for the task. This untangling can also be an emergent property, meaning the model also learns to untangle some types of object manifolds without being trained to do so explicitly.

We present several key findings:

1. We find significant untangling of word manifolds in different model architectures trained on speech tasks. We also see emergent untangling of higher-level concepts such as words, phonemes, and parts of speech in an end-to-end ASR model (Deep Speech 2).

2. Both a CNN architecture and the end-to-end ASR model converge on remarkably similar behavior despite being trained for different tasks and built with different computational blocks. They both learn to discard nuisance acoustic variations, and exhibit untangling for task relevant information.

3. Temporal dynamics in recurrent layers reveal untangling over recurrent time steps, in the form of smaller manifold radius, lower manifold dimensionality.

In addition, we show the generality of auditory untangling with speaker manifolds in a network trained on a speaker recognition task, that are not evident in either the end-to-end ASR model or the model trained explicitly to recognize words. These results provide the first geometric evidence for untangling of manifolds, from phonemes to parts-of-speech, in deep neural networks for speech recognition.

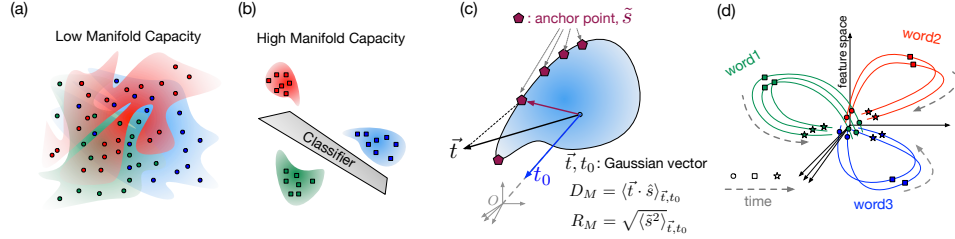

Figure 1: **Illustration of word manifolds.** (a) highly tangled manifolds, in low capacity regime (b) untangled manifolds, in high capacity regime (c) Manifold Dimension captures the projection of a Gaussian vector onto the direction of an anchor point, and Manifold Radius captures the norm of an anchor point in manifold subspace. (d) Illustration of untanglement of words over time.

## 2 Methods

To understand representation in speech models, we first train a neural network on a corpus of transcribed speech. Then, we use the trained models to extract per-layer representations at every time step on each corpus stimulus. Finally, we apply the mean-field theoretic manifold analysis technique [24, 25] (hereafter, MFTMA technique) to measure manifold capacity and other manifold geometric properties (radius, dimension, correlation) on a subsample of the test dataset.

Formally, if we have $P$ objects (e.g. words), we construct a dataset $\mathcal{D}$ with pairs $(x_i, y_i)$, where $x_i$ is the auditory input, and $y_i \in \{1, 2, ..., P\}$ is the object class. Given a neural network $N(x)$, we extract $N_t^l(x)$, which is the output of the network at time $t$ in layer $l$, for all inputs $x$ whose corresponding label is $p$, for each $p \in \{1, 2, ..., P\}$. The object manifold at layer $l$ for class $p$ is then defined as the point cloud of activations obtained from the different examples $x_i$ of the $p^{th}$ class. We then apply the MFTMA technique to this set of activations to compute the manifold capacity, manifold dimension, radius, and correlations for that manifold.

The manifold capacity obtained by the MFTMA technique captures the linear separability of object manifolds. Furthermore, as shown in [24, 25] and outlined in SM Sect. 1.2, the mean-field theory calculation of manifold capacity also gives a concrete connection between the measure of separability and the size and dimensionality of the manifolds. This analysis therefore gives additional insight into both the separability of object manifolds, and how this separability is achieved geometrically.

We measure these properties under different manifold types, including categories such as phonemes and words, or linguistic feature categories such as part-of-speech tags. This allows us to quantify the amount of invariant object information and the characteristics of the emergent geometry in the representations learned by the speech models.

### 2.1 Object Manifold Capacity and the mean-field theoretic manifold analysis (MFTMA)

In a system where $P$ object manifolds are represented by $N$ features, the 'load' in the system is defined by $\alpha = P/N$. When $\alpha$ is small, i.e. a small number of object manifolds are embedded in a high dimensional feature space, it's easy to find a separating hyperplane for a random dichotomy[2] of the manifolds. When $\alpha$ is large, too many categories are squeezed in a low dimensional feature space, rendering the manifolds highly inseparable. *Manifold capacity* refers to the critical load, $\alpha_C = P/N$, defined by the critical number of object manifolds, $P$, that can be linearly separated given $N$ features. Above $\alpha_C$, most dichotomies are inseparable, and below $\alpha_C$, most are separable[24, 25]. This framework generalizes the notion of the perceptron storage capacity [26] from points to manifolds, re-defining the unit of counting to be object manifolds rather than individual points. The manifold capacity thus serves as a measure of the linearly decodable information about object identity per feature, and it can be measured from data in two ways:

1. **Empirical Manifold Capacity,** $\alpha_{SIM}$: the manifold capacity can be measured empirically with a bisection search to find the critical number of features $N$ such that the fraction of linearly separable random dichotomies is close to $1/2$.

2. **Mean Field Theoretic Manifold Capacity,** $\alpha_{MFT}$: can be estimated using the replica mean field formalism with the framework introduced by [24, 25]. $\alpha_{MFT}$ is estimated from the statistics of *anchor points* (shown in Fig. 1(c)), $\tilde{s}$, a representative point for a linear classification[3].

The manifold capacity for point-cloud manifolds is lower bounded by the case where there is no manifold structure. This lower bound is given by [29, 24]. In this work, we show $\alpha_{MFT}/\alpha_{LB}$ for a comparison between datasets of different sizes and therefore different lower bounds.

Manifold capacity is closely related to the underlying geometric properties of the object manifolds. Recent work demonstrates that the manifold classification capacity can be predicted by an object manifold's *Manifold Dimension*, $D_M$, *Manifold Radius*, $R_M$, and the correlations between the centroids of the manifolds [23, 24, 25]. These geometrical properties capture the statistical properties of the anchor points, the representative support vectors of each manifold relevant for the linear classification, which change as the choice of other manifolds vary [24]. The MFTMA technique also measures these quantities, along with the manifold capacity:

**Manifold Dimension,** $D_M$: $D_M$ captures the dimensions realized by the anchor point from the guiding Gaussian vectors shown in Fig. 1(c), and estimates the average embedding dimension of the manifold contributing to the classification. This is upper bounded by $\min(M, N)$, where $M$ is the number of points per each manifold, and $N$, the feature dimension. In this work, $M < N$, and we present $D_M/M$ for fair comparison between different datasets.

**Manifold Radius,** $R_M$: $R_M$ is the average distance between the manifold center and the anchor points as shown in Fig. 1(c). Note that $R_M$ is the size relative to the norm of the manifold center, reflecting the fact that the relative scale of the manifold compared to the overall distribution is what matters for linear separability, rather than the absolute scale.

**Center Correlations,** $\rho_{center}$: $\rho_{center}$ measures how correlated the locations of these object manifolds are, and is calculated as the average of pairwise correlations between manifold centers. [25].

It has been suggested that the capacity is inversely correlated with $D_M$, $R_M$, and center correlation [24, 25]. Details for computing anchor points, $\tilde{s}$ can be found in the description of the mean-field theoretic manifold capacity algorithm, and the summary of the method is provided in the SM.

In addition to mean-field theoretic manifold properties, we measure the dimensionality of the data across different categories with two popular measures for dimensionality of complex datasets:

**Participation Ratio** $D_{PR}$: $D_{PR}$ is defined as $\frac{(\sum_i \lambda_i)^2}{\sum_i \lambda_i^2}$, where $\lambda_i$ is the $i^{th}$ eigenvalue of the covariance of the data, measuring how many dimensions of the eigen-spectrum are active [17].

**Explained Variance Dimension** $D_{EV}$: $D_{EV}$ is defined as the number of principal components required to explain a fixed percentage (90% in this paper) of the total variance [30].

Note that in all of the dimensional measures, i.e. $D_M$, $D_{PR}$ and $D_{EV}$ are upper bounded by the number of eigenvalues, which is $\min(\#samples, \#features)$ (these values are used in (Fig. 2-6)).

## 2.2 Models and datasets

We examined two speech recognition models. The first model is a CNN model based on [14], with a small modification to use batch normalization layers instead of local response normalization (full architecture can be found in Table SM1). We trained the model on two tasks: word recognition and speaker recognition. For word recognition, we trained on two second segments from a combination of the WSJ Corpus [31] and Spoken Wikipedia Corpora [32], with noise augmentation from AudioSet backgrounds [33]. For more training details, please see the SM.

The second is an end-to-end ASR model, Deep Speech 2 (DS2) [5], based on an open source implementation[4]. DS2 is trained to produce accurate character-level output with the Connectionist Temporal Classification (CTC) loss function [34]. The full architecture can be found in Table SM2. Our model was trained on the 960 hour training portion of the LibriSpeech dataset [35], achieving a

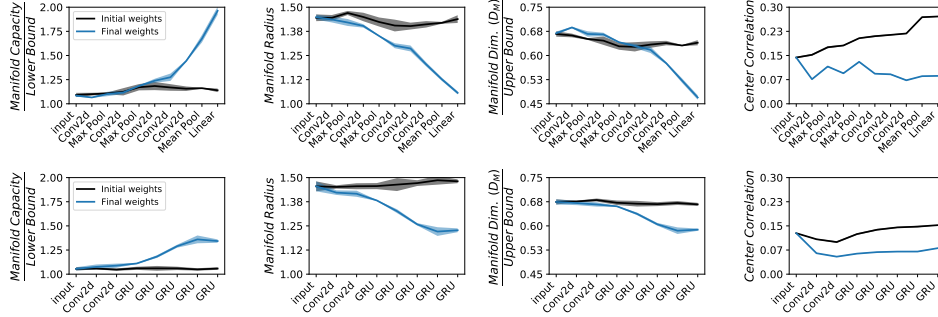

Figure 2: **Word manifold capacity emerges in both the CNN word classification model and the end to end ASR model (DS2).** Top: As expected, CNN model trained with explicit word supervision (blue lines) exhibits strong capacity in later layers, compared to the initial weights (black lines). This increase is due to reduced radius and dimension, as well as decorrelation. Bottom: A similar trend emerges in DS2 without training with explicit word supervision. In both, capacity is normalized against the theoretical lower bound (See Methods). The shaded area represents 95% confidence interval hereafter.

word error rate (WER) of 12%, and 22.7% respectively on the clean and other partitions of the test set without the use of a language model. The model trained on LibriSpeech also performs reasonably well on the TIMIT dataset, with a WER of 29.9% without using a language model.

We followed the procedure described in Sec. 2 to the construct manifold datasets for several different types of object categories using holdout data not seen by the models during training. The datasets used in the analysis of each model were as follows:

**CNN manifolds datasets**: Word manifolds from the CNN dataset were measured using data from the WSJ corpus. Each of the $P = 50$ word manifolds consist of $M = 50$ speakers saying the word, and each of the $P = 50$ speaker manifolds consist of one speaker saying $M = 50$ different words.

**DS2 manifolds datasets**:[5] Word and speaker manifolds were taken from the test portion of the LibriSpeech dataset. For word manifolds, $P = 50$ words with $M = 20$ examples each were selected, ensuring each example came from a different speaker. For speaker manifolds $P = 50$ speakers were selected with $M = 20$ utterances per speaker.

For the comparison between character, phoneme, word, and parts-of-speech manifolds, similar manifold datasets were also constructed from TIMIT, which includes phoneme and word alignment. $P = 50$ and $M = 50$ were used for character and phoneme manifolds, but owing to the smaller size of TIMIT, $P = 23$ and $M = 20$ were used for word manifolds. Likewise, we used a set of $P = 22$ tags, with $M = 50$ for the parts-of speech manifolds.

**Feature Extraction** For each layer of the CNN and DS2 models, the activations were measured for each exemplar and 5000 random projections with unit length were computed on which to measure the geometric properties. For temporal analysis in the recurrent DS2 model, full features were extracted for each time step.

## 3 Results

### 3.1 Untangling of words

We first investigated the CNN model, which was trained to identify words from a fixed vocabulary using the dataset described in 2.2. Since this model had explicit word level supervision, we observed that the word classes were more separable (higher capacity) over the course of the network layers (Figure 2 (Top) as expected. This word manifold capacity was not observed with the initial weights of the model (Figure 2, black lines). As a negative control, we trained the same CNN architecture

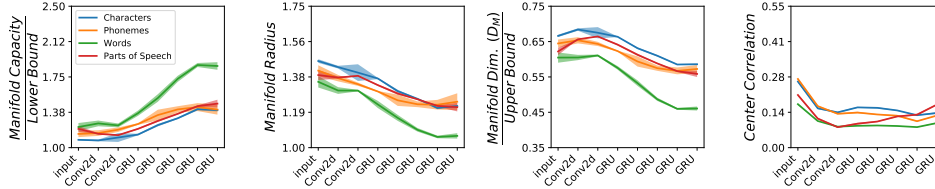

Figure 3: **Character, phoneme, word, and part-of-speech manifolds emerge in the DS2 model.**

to identify speakers instead, and did not observe increased word manifold capacity after training (See Fig. SM3). These results demonstrate that word separability is task dependent and not a trivial consequence of the input, acoustic features, or model architecture. Furthermore, the MFTMA metrics reveal that this increased word capacity in later layers is due to both a reduction in the manifold radius and the manifold dimension.

Most end-to-end ASR systems are not trained to explicitly classify words in the input, owing to the difficulty in collecting and annotating large datasets with word level alignment, and the large vocabulary size of natural speech. Instead, models such as Deep Speech 2 are trained to output character-level sequences. Despite not being trained to explicitly classify words, the untangling of words was also emergent on the LibriSpeech dataset (Figure 2, bottom). In both CNN and DS2 models, we present the capacity values normalized by the lower bound and manifold dimension is normalized by the upper bound (Sec. 2.1), for easy comparison.

Surprisingly, across the CNN and recurrent DS2 architectures, the trend in the other geometric metrics were similar. The manifold capacities improve in downstream layers, and the reduction in manifold dimension and radius similarly occurs in downstream layers. Interestingly, word manifolds increases dramatically in the last layer of CNN, but only modestly in the last layers of DS2, perhaps owing to the fact that CNN model here is explicitly trained on word labels, while in the DS2, the word manifolds are emergent properties. Notably, the random weights of the initial model increase correlation across the layers in both networks, but the training significantly decreases center correlation in both models. More analysis on training epochs are given in Section 3.4 and in Sect. SM4.5.

## 3.2 Untangling of other speech objects

In addition to words, it is possible that auditory objects at different levels of abstraction are also untangled in the ASR task. To investigate this, we look for evidence of untangling at four levels of abstraction: characters, phonemes, words (each word class contains multiple phonemes and multiple characters), and part of speech tags (each part of speech class contains multiple words). These experiments were done on the end-to-end ASR model (DS2). Results for these four object types are shown in Fig. 3.

On the surface, we see all four of these quantities becoming untangled to some degree as the inputs are propagated through the network. However, in the final layers of the network, the untangling of words is far more prominent than that of phonemes (see the numerical values of capacity in Fig. 3). This suggests that the speech models may need to abstract over the phonetic information and character information (i.e. silent letters in words). While the higher capacity is due to a lower manifold radius and dimension, we note that the manifold dimension is significantly lower for words than it is for phonemes and parts of speech. The phoneme capacity starts to increase only after the second convolution layer, consistent with prior findings [6].

## 3.3 Loss of speaker information

In addition to learning to increase the separability of information relevant to the ASR task, robust ASR models should also learn to ignore unimportant differences in their inputs. For example, different instances of the same word spoken by different speakers must be recognized as the same word in spite of the variations in pronunciation, rate of speech, etc. If this is true, we expect that the separability of different utterances from the same speaker is decreased by the network, or at least not increased, as this information is not relevant to the ASR task.

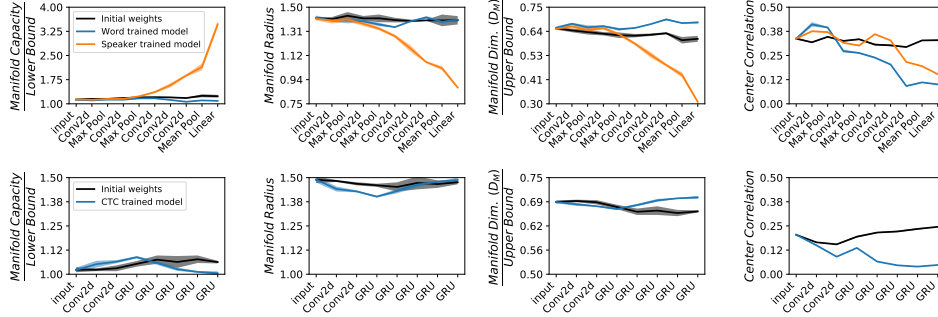

Figure 4: **Speaker Manifolds disappear in different speech models.** Top: CNN ((black) before, and initial weights, trained on (blue) words. and (orange) speakers); Bottom: DS2 ((black) initial weights, (blue) trained on ASR task).

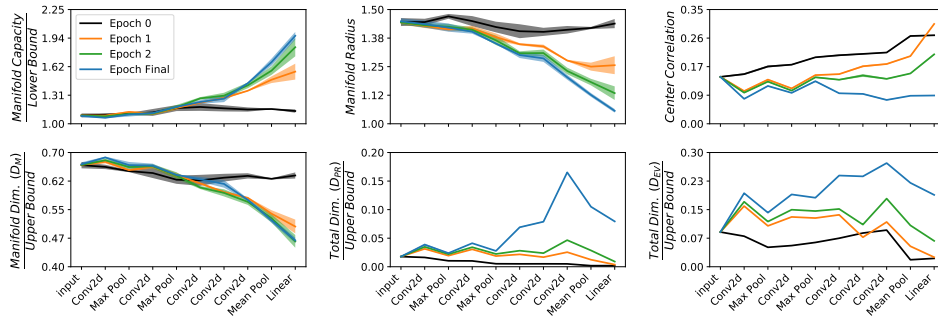

Figure 5: **Evolution of word manifolds via epochs of training, CNN.** Manifold capacity improves over epochs of training, while manifold dimension, radius, correlations decrease over training. Total data dimension ($D_{PR}$, $D_{EV}$) is inversely correlated with center correlation and increases over training epochs. Similar trends are observed in the DS2 model (SM).

Indeed, Figure 4 (Top) shows that for the CNN model trained to classify words, the separability of speaker manifolds defined by the dataset described in 2.2 decreases deeper in the network, and is even lower than in the untrained network. In contrast, when training a CNN explicitly to do speaker recognition the separability of speaker manifolds increases in later layers (while the separability of word manifolds does not, see Sect. SM4.2), demonstrating that the lack of speaker separation and the presence of word information is due to the specific task being optimized.

A similar trend also appears in the DS2 model, as shown in Fig. 4 (Bottom). In both the CNN trained to recognize words and DS2 models, speaker manifolds become more tangled after training, and in both cases we see that this happens due to an increase in the dimensionality of the speaker manifolds, as the manifold radius remains unchanged after training, and the center correlation decreases. In some sense, this mirrors the results in Sec. 3.1 and Sec. 3.2 where the model untangles word level information by decreasing the manifold dimension, and here discards information by increasing the manifold dimension instead.

For the CNN model, the decrease in speaker separability occurs mostly uniformly across the layers, where in DS2, the separability only drops in the last half of the network in the recurrent layers. Surprisingly, the early convolution layers of the DS2 model show either unchanged or slightly increased separability of speaker manifolds. We note that the decrease in speaker separability coincides with both the increase in total dimension as measured by participation ratio and explained variance, as seen in Fig. 5 and Fig. SM6, as well as a decrease in center correlation.

### 3.4 Trends over training epochs

In addition to evaluating the MFTMA quantities and effective dimension on fully trained networks, the analysis can also be done as the training progresses. Figure 5 shows the early stages of training

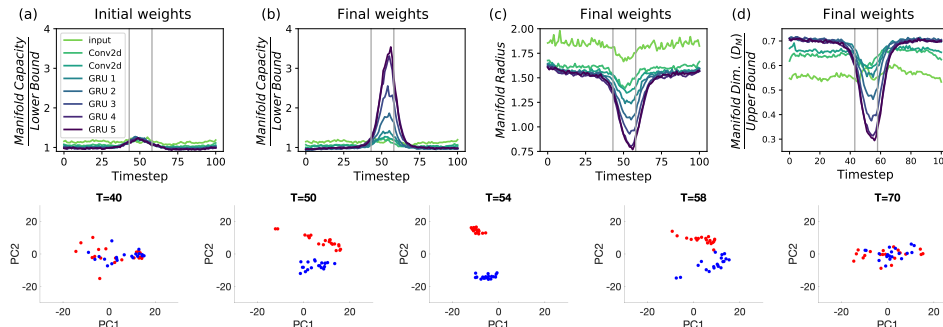

Figure 6: **Untangling of word manifolds in input timesteps.** (Top) Evolution of Librispeech word manifolds in timesteps, DS2 model (hypothesized in Fig 1). (a) Epoch 0 model, capacity (b-d) fully trained model, (b) capacity, (c) manifold radius, (d) manifold dimension. Vertical lines show the average word boundaries. (Bottom) Untanglement of two words over timesteps (T=40 to 70) in GRU 5 layer of DS2, projected to 2 PCs.

of the word recognition CNN model (See Fig. SM6 for the early stages of training for DS2, which shows similar trends). The capacity, manifold dimension, manifold radius, and center correlations quickly converge to those measured on the final epochs. Interestingly, the total data dimension (measured by $D_{PR}$, and $D_{EV}$) increases with training epochs unlike the manifold dimension, $D_M$, which decreases over training in both models. Intuitively, the training procedure tries to embed different categories in different directions while compressing them, resulting in a lower $D_M$ and a lower center correlation. The increase in total $D_{PR}$ could be related to lowered manifold center correlation. The total dimension $D_{PR}$ could play the role of the larger 'effective' ambient dimension, in turn improving linear separability [36].

## 3.5 Untanglement of words over time

The above experiments were performed without considering the sequential nature of the inputs. Here, we compute these measures of untangling on each time step separately. This approach can interrogate the role of time in the computation, especially in recurrent models processing arbitrary length inputs.

Figure 6 shows the behavior of capacity, manifold radius, and manifold dimension over the different time steps in the recurrent layers of the end-to-end ASR model (DS2) for the word inputs used in Sec. 3.1. As is perhaps expected, the separability is at the theoretical lower bound for times far away from the word of interest, and peaks near the location of the word. This behavior arises due to the decrease in radius and dimension. However, the peak does not occur at the center of the word, owing to the arbitrary time alignment in the CTC cost function, as noted in [37]. Do the inputs far in time from the word window play a significant role in the untangling of words? While we omit further investigation of this here due to space constraints, an experiment on varying length inputs can be found in the SM.

Interestingly, the capacity measured at each input time step has a peak relative capacity of 3.6 (Fig. 6), much larger than the capacity measured from a random projection across all features, at 1.4 (Fig. 2). This is despite the lower feature dimension in the time step analysis (due to considering the features at only one time step, rather than the full activations). This implies that sequential processing also massages the representation in a meaningful way, such that a snapshot at a peak time step has a well separated, compressed representation, captured by the small value of $D_M$ and $R_M$. Analogous to 1(d), the efficiency of temporal separation is illustrated in Fig. 6, bottom.

## 4 Conclusion

In this paper we studied the emergent geometric properties of speech objects and their linear separability, measured by manifold capacity. Across different networks and datasets, we find that linear separability of auditory class objects improves across the deep network layers, consistent with the untangling hypothesis in vision [2]. Word manifold's capacity arises across the deep layers, due to

emergent geometric properties, reducing manifold dimension, radius and center correlations. Characterization of manifolds across training epochs suggests that word untangling is a result of training, as random weights do not untangle word information in the CNN or DS2. As representations in ASR systems evolve on the timescale of input sequences, we find that separation between different words emerges temporally, by measuring capacity and geometric properties at every frame.

Speech data naturally has auditory objects of different scales embedded in the same sequence of sound and we observed here an emergence and untangling of other speech objects such as phonemes, and part-of-speech manifolds. Interestingly, speaker information (measured by speaker manifolds) dissipates across layers. This effect is due to the network being trained for word classification, since CNNs trained with speaker classification are observed to untangle speaker manifolds. Interestingly, the transfer between speaker ID and word recognition was not very good, and a network trained for both speaker ID and words showed emergence of both manifolds, but these tasks were not synergistic (see Fig. SM4). These results suggest that the task transfer performance is closely related to the task structure, and can be captured by representation geometry.

Our methodology and results suggest many interesting future directions. Among other things, we hope that our work will motivate: (1) the theory-driven geometric analysis of representation untangling in tasks with temporal structure; (2) the search for the mechanistic relation between the network architecture, learned parameters, and structure of the stimuli via the lens of geometry; (3) the future study of competing vs. synergistic tasks enabled by the powerful geometric analysis tool.

**Acknowledgments**

We thank Yonatan Belinkov, Haim Sompolinsky, Larry Abbott, Tyler Lee, Anthony Ndirango, Gokce Keskin, and Ting Gong for helpful discussions. This work was funded by NSF grant BCS-1634050 to J.H.M. and a DOE CSGF Fellowship to J.J.F. S.C acknowledges support by Intel Corporate Research Grant, NSF NeuroNex Award DBI-1707398, and The Gatsby Charitable Foundation.

## Footnotes

[1]Our implementation of the analysis methods: https://github.com/schung039/neural_manifolds_replicaMFT

[2]Here, we define a random dichotomy as an assignment of random $\pm 1$ labels to each manifold

[3]See SM for exact relationship between $\tilde{s}$ and capacity, the outline of the code, and a demonstration that MFT manifold capacity matches the empirical capacity (given in Fig. SM1)

[4]https://github.com/SeanNaren/deepspeech.pytorch

[5]See Sect. SM3.2 for more details on the construction and composition of the datasets used for experiments on this model.

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
