[Supplementary Material]

# Untangling in Invariant Speech Recognition - Supplemental Material

## 1 Details on measuring empirical and theoretical manifold capacity

### 1.1 Empirical Manifold Capacity

Here we provide a detailed description for empirically finding a manifold capacity, for a given number of object class manifolds , $P$, by finding a critical number of feature dimensions, $N_c$, such that the fraction of separable dichotomies of random assignment of +/- labels to a given manifolds is at 0.5 on average (Fig. SM1). If the feature dimension $N$ is larger than critical $N_c$, the fraction of separable dichotomies will be close to 1 (hence, the system is in a linearly separable regime, Fig. SM1). If the feature dimension $N$ is smaller than critical $N_c$, the fraction of separable dichotomies will be close to 0 (the system is in a linearly in-separable regime, Fig. SM1). The algorithm finds $N_c$ by doing a bisection search on N, such that "fraction of linearly separable dichotomies" for $N_c$ is 0.5, midpoint between 1 (separable) and 0 (inseparable) on average. At the cricical $N_c$, the capacity is defined to be $P/N_c$. In our experiments shown in Fig. SM1, we used randomly sampled 101 dichotomies, to compute fraction of linear separability.

Figure SM1: **Measured capacity matches theoretical prediction.** (Left) Across multiple datasets (TIMIT, Librispeech) and manifolds (words, speakers, phonemes, part-of-speech tags) for the DS2 model, the measured capacity closely matches the theoretical capacity. Dotted line indicates unity. (Right) Empirical capacity is measured by a bisection search for critical N s.t. fraction of separable datasets cross 0.5

### 1.2 Mean-Field Theoretic (MFT) Manifold Capacity and Geometry

Here, we provide a summmary for finding a theoretical estimation of manifold capacity using mean-field theoretic approach. It has been proven that the general form of the inverse MFT capacity, exact in the thermodynamic limit, is given by:

$$\alpha_{MFT}^{-1} = \left\langle \frac{\left[t_0 + \vec{t} \cdot \tilde{s}(\vec{t})\right]_+^2}{1 + \left\| \tilde{s}(\vec{t}) \right\|^2} \right\rangle_{\vec{t}, t_0}$$

where $\langle \ldots \rangle_{\vec{t}, t_0}$ is an average over random $D$- and 1- dimensional vectors $\vec{t}, t_0$ whose components are i.i.d. normally distributed $t_i \sim \mathcal{N}(0, 1)$.

Central to this framework is the notion of *anchor points*, $\tilde{s}$ (section 2.1 in the main text), uniquely given by each $\vec{t}, t_0$, representing contributions from all other object manifolds, in their random orientations. For each $\vec{t}, t_0$, $\tilde{s}$ is uniquely defined as a subgradient that obeys the KKT conditions, hence, $\tilde{s}$ in KKT interpretation, represents a weighted sum of support vectors contributing to the linearly separating solution.

These anchor points play a key role in estimating the manifold's geometric properties, given as: $R_{\mathrm{M}}^2 = \left\langle \left\| \tilde{s}(\vec{T}) \right\|^2 \right\rangle_{\vec{T}}$ and $D_{\mathrm{M}} = \left\langle \left( \vec{t} \cdot \hat{s}(\vec{T}) \right)^2 \right\rangle_{\vec{T}}$ where $\hat{s}$ is a unit vector in the direction of $\tilde{s}$, and $\vec{T} = (\vec{t}, t_0)$, which is a combined coordinate for manifold's embedded space, and manifold's center direction (in general, if we compute the geometric properties in the ambient dimension, it includes both the embedded space and center direction).

The manifold dimension measures the dimensional spread between $\vec{t}$ and its unique anchor point $\tilde{s}$ in $D$ dimensions (the coordinates in which each manifold is embedded).

In high dimension, the geometric properties predict the MFT manifold capacity, by

$$\alpha_{\mathrm{MFT}} \approx \alpha_{\mathrm{Ball}} \left( R_{\mathrm{M}}, \, D_{\mathrm{M}} \right) \tag{1}$$

where,

$$\alpha_{\mathrm{Ball}}^{-1}(R, D) = \int_{-\infty}^{R\sqrt{D}} Dt_0 \frac{(R\sqrt{D} - t_0)^2}{R^2 + 1} \tag{2}$$

Note that the above formalism is assuming that the manifolds are in random locations and orientations, and in real data, the manifolds have various correlations. So, we apply the above formalism onto the data that has been projected into the null spaces of centers, using the method proposed by [1].

The validity of this method is shown in Fig. SM1, where we demonstrate the good match between the empirical manifold capacity (computed using a method in Section. 1.1) and the mean-field theoretical estimation of manifold capacity (using the algorithm provided in this section).

For more details on the theoretical derivations and interpretations for the mean-field theoretic algorithm, see [1][2].

---

**Algorithm 1** compute_geometric_properties: Mean-field theoretic capacity and geometry

---

**Function** compute_geometric_properties($\{X^\mu\}$)

**Input**: Categorical data $\left\{ X_i^\mu \in \mathbb{R}^N \right\}_{i \in [1..M_\mu]}^{\mu=1..P}$ ($P$=#Manifolds, $M_\mu$=#Samples per $\mu$th Manifold)

1. Subtract global mean and update $\left\{ X_i^\mu \in \mathbb{R}^N \right\}_{i \in [1..M_\mu]}^{\mu=1..P}$

2. Compute centers of each manifold $\{ \vec{c}^\mu \in \mathbb{R}^N \}^{\mu=1,...,P}$

3. Compute center correlations $\delta_{CC}$

4. Find subspace shared by manifold centers* : $\{C^\mu\} = \mathrm{find\_center\_subspace} \left( \{X^\mu\} \right)$

5. Project original data into null space of center subspaces*: $\{X^{\perp\mu}\} = \mathrm{find\_residual\_data} \left( \{X^\mu, C^\mu\} \right)$

6. Normalize data s.t. center norms are 1**: $X^{0\perp\mu} = \mathrm{manifold\_normalize} \left( X^{\perp\mu} \right)$

7. For $\mu = 1..P$, calculate geometry**: $D_M^\mu, R_M^\mu, \alpha_c^\mu = \mathrm{manifold\_geometry} \left( X^{0\perp\mu} \right)$

**Output**: $\{D_M^\mu\}_{\mu=1}^P$, $\{R_M^\mu\}_{\mu=1}^P$, $\{\alpha_c^\mu\}_{\mu=1}^P$

---

* is based on [1], and ** is based on [2].

## 2 Details of models used in experiments

### 2.1 CNN Model

Table 1: Word (and Speaker) CNN Model Architecture

| Layer | Type | Size |
|---|---|---|
| 0 | Input | $256 \times 256$ cochleagram |
| 1 | 2D Convolution | 96 filters of shape $9 \times 9$, stride 3 |
| 2 | ReLu | - |
| 3 | MaxPool | window $3 \times 3$, stride 2 |
| 4 | 2D BatchNorm | - |
| 5 | 2D Convolution | 256 filters of shape $5 \times 5$, stride 2 |
| 6 | ReLu | - |
| 7 | MaxPool | window $3 \times 3$, stride 2 |
| 8 | 2D BatchNorm | - |
| 9 | 2D Convolution | 512 filters of shape $3 \times 3$, stride 1 |
| 10 | ReLu | - |
| 11 | 2D Convolution | 1024 filters of shape $3 \times 3$, stride 1 |
| 12 | ReLu | - |
| 13 | 2D Convolution | 512 filters of shape $3 \times 3$, stride 1 |
| 14 | ReLu | - |
| 15 | AveragePool | window $3 \times 3$, stride 2 |
| 16 | Linear | 4096 units |
| 17 | ReLu | - |
| 18 | Dropout | 0.5 prob during training |
| 19 | Linear | Num Classes |

For word recognition, we trained on two second segments from a combination of the WSJ Corpus [3] and Spoken Wikipedia Corpus [4], with noise augmentation from audio set backgrounds [5]. We selected two second sound segments such that a single word occurs at one second. For the training set, we selected words and speaker classes such that each class contained 50 unique cross class labels (ie 50 unique speakers had to say each of the word classes). We also selected words and speaker classes that each contained at least 200 unique utterances, and ensured that each category could contain a maximum of 25% of a single cross category label (ie for a given word class, a maximum of 25% of the utterances could come from a single speaker), the maximum number of utterances in any word category was less than 2000, and the maximum number of utterances within any speaker category was less than 2000. Data augmentation during training consisted jittering the input in time and placing the exemplars on different audioset backgrounds.

The resulting training dataset contained 230,357 segments in 433 speaker classes and 793 word classes. The word recognition network achieved a WER of 22.7% on the test set, and the speaker recognition network achieved an error rate of 1% on the test set.

### 2.2 Deep Speech 2 Model

The ASR model used in experiments is based on the Deep Speech 2 architecture[6]. A complete specification of the model is given in Table 2. The version we used is based on a popular open source implementation available at https://github.com/SeanNaren/deepspeech.pytorch

Table 2: End-to-end ASR model architecture

| Layer | Type | Size |
|-------|------|------|
| 0 | Input | $T \times 161$ spectral features |
| 1 | 2D Convolution | 32 filters of shape $41 \times 11$, stride 2 |
| 2 | 2D BatchNorm | - |
| 3 | HardTanh | - |
| 4 | 2D Convolution | 32 filters of shape $21 \times 11$, stride 2 in time only |
| 5 | 2D BatchNorm | - |
| 6 | HardTanh | - |
| 7 | Bidirectional GRU | 800 |
| 8 | 1D BatchNorm[6] | - |
| 9 | Bidirectional GRU | 800 |
| 10 | 1D BatchNorm[6] | - |
| 11 | Bidirectional GRU | 800 |
| 12 | 1D BatchNorm[6] | - |
| 13 | Bidirectional GRU | 800 |
| 14 | 1D BatchNorm[6] | - |
| 15 | Bidirectional GRU | 800 |
| 16 | 1D BatchNorm[6] | - |
| 17 | Linear | $800 \times 29$ |

This model was trained on the 960 hour training portion of the LibriSpeech dataset [7] for 68 epochs with an initial learning rate of 0.0003 and a learning rate annealing of 1.1. The trained model has a word error rate (WER) of 12%, 22.7% respectively on the clean and other partitions of the test set without the use of a language model. The WER for different training epochs is shown in Figure **??**. The model also performs reasonably well on the TIMIT dataset, with a WER of 29.9% without using a language model.

# 3 Details on the manifold datasets

## 3.1 Manifold datasets used for experiments on the CNN Model

The words used in the CNN model word manifold dataset were selected from the WSJ corpus. The selected words were: whether, likely, large, found, declined, common, these, number, after, through, don't, workers, began, street, during, other, agreement, management, trying, office, often, members, local, without, before, continue, money, personal, first, almost, didn't, enough, results, about, their, party, would, could, people, should, military, former, january, selling, reported, times, called, holding, small, increased.

The midpoint of each word was used as the midpoint of 2s of audio, such that the amount of context was equal before and after the word of interest.

## 3.2 Manifold datasets used for experiments on the DS2 model

### 3.2.1 Words from LibriSpeech

Words were selected from a subset of the dev-clean (holdout data) portion of the LibriSpeech corpus. Word boundaries were found using forced alignment[1]. Words were selected according to the following criteria:

- Words must be at least four characters long

- Words must occur in the center of at least two seconds of audio

- Words must have at least 20 examples each from different speakers.

The resulting dataset consists of 50 different words, with 20 examples of each. The selected words were: made, have, while, still, being, upon, said, much, into, never, good, eyes, what, little, which, more, away, before, back, always, himself, like, from, down, such, when, under, other, should, after, another, thought, would, time, same, first, about, than, think, even, only, make, great, most, might, where, both, know, life, through.

The resulting examples were two seconds long, with the center of the word occurring at the midpoint of the signal as in the CNN word manifolds dataset. The lower bound on capacity in this experiment is $0.1 \leq \alpha$ as there are $M = 20$ samples per word.

### 3.2.2   Words from TIMIT

Words from TIMIT were selected according to the same criteria as the LibriSpeech words, with the exceptions that the input length was reduced to $1s$, to ensure enough samples could be obtained, and the number of samples per word was increased to $50$. This resulted in the selection of 23 words. The selected words were: carry, dark, every, from, greasy, have, into, like, more, oily, suit, that, their, they, this, through, wash, water, were, will, with, would, your.

We used the same alignment as in the CNN word and LibriSpeech word manifolds dataset, with the exception of the shorter $1s$ overall length. The lower bound on capacity in this experiment is $0.04 \leq \alpha$ as there are $M = 50$ samples per word.

### 3.2.3   Phonemes from TIMIT

Phonemes were selected from the TIMIT dataset according to the following criteria:

- Phonemes must occur in the center of at least one second of audio
- Phonemes must have at least 50 examples each from different speakers

The resulting dataset consists of $P = 59$ phonemes, with $M = 50$ examples each. The selected phonemes are aa, ao, ax-h, b, d, dx, em, er, g, hv, iy, kcl, n, ow, pau, q, sh, th, ux, y, ae, aw, axr, bcl, dcl, eh, en, ey, gcl, ih, jh, l, ng, oy, pcl, r, t, uh, v, z, ah, ax, ay, ch, dh, el, epi, f, hh, ix, k, m, nx, p, s, tcl, uw, w, zh.

The center of the phoneme (defined as the midpoint between the boundaries) was used as the center of $1s$ of audio, similar to the construction of the word manifolds datasets. The lower bound on capacity in this experiment is $0.04 \leq \alpha$ as there are $M = 50$ samples per phoneme.

## 4   Additional experiments

### 4.1   Evolution of word error rate (WER) and manifold capacity during training

Figure SM2: **WER vs. word manifold capacity** In both the CNN (left) and DS2 (right) models, the decrease in WER during training occurs simultaneously with the rise in word manifold capacity in the final nonlinear layer. Each data points represents a different epoch of training.

During training, the word error rate (WER) decreases for both the CNN (when trained to recognize words) and DS2. At the same time, the separability of words in the later layers of both networks increases as measured by word manifold capacity. Figure SM2 shows that in the final nonlinear layer of the networks, the change in these two quantities occurs simultaneously.

## 4.2 Word manifolds on Speaker recognition CNN

Figure SM3: **Word manifolds disappear in a model trained on a speaker recognition task** Here, the CNN model was trained to recognize individual speakers, resulting in no untanglement of word manifolds in contrast to the word trained model.

As a further negative control, we also trained the CNN model on a speaker identification task, and measured the MFTMA metrics on both speaker and word manifolds. In this scenario, we see untanglement of speaker manifolds. However, we do not see untanglement of word manifolds, which are more tangled in the final layers than with random (initial) weights of the network.

## 4.3 Word and speaker manifolds on a multitask trained CNN

Figure SM4: **Word and speaker manifolds in a model trained simultaneously on word recognition and speaker recognition tasks**. While word and speaker manifolds emerge, these two tasks are not synergistic

When trained on both the word recognition and speaker identification tasks, the MFTMA metrics reveal some untangling of both word and speaker manifolds as shown in Fig. SM4. However, the amount of untanglement is significantly lower than when the model is trained on a single task, indicating that the two tasks are not synergistic.

## 4.4 Analysis across time in the DS2 model before training

Figure SM5: **Untangling of word manifolds in input timesteps Before training DS2.** No improvement over the layers, and slightly more information in the midpoint.

When the analysis is run over timesteps in DS2, we do not see evidence for increasing untanglement of word manifolds with depth before the network is trained as shown in Fig. SM5. While we do see a slight increase in capacity along with a corresponding decrease in manifold radius an dimension near

the location of the center words, these metrics remain constant through the layers of the network in contrast to the behavior seen after the network has been trained.

## 4.5 Analysis on DS2 training epochs

Figure SM6: **Evolution of word manifolds via epochs of training, DS2** Manifold capacity improves over epochs of training, while manifold dimension, radius, correlations decrease over training. Total data dimension ($D_{PR}$, $D_{EV}$) is inversely correlated with center correlation and increases over training epochs. Error bars show 95% confidence intervals of the mean.

Figure SM6 shows the MFTMA metrics over the course of training DS2. As a single epoch is quite large (960 hours of audio) we plot the metrics at two points within the first epoch. Here, the trends are similar to those observed in the CNN model in that the manifold dimension shrinks, while the total data dimension as measured by $D_{PR}$ and $D_{EV}$ rises. As in the CNN model, the manifolds are compressed (smaller manifold radius, dimension) while their centers become more spread out (larger $D_{PR}$, $D_{EV}$).

## 4.6 Experiment on DS2 with different length contexts

Figure SM7: **Experiment on varying length contexts (Librispeech words)**

We also carried out an exploratory analysis on the role of context in the recognition of words in the DS2 model. Figure SM7 shows the results of the MFTMA on word manifolds with different length context windows. The results shown here were measured at the center time step. The manifold datasets were constructed by taking the LibriSpeech word manifolds dataset (consisting of words centered in 2s of audio) and reducing the context symmetrically on either side of the center word. While some differences do emerge for shorter contexts in the final two layers of the network, indicating that perhaps context is important in these layers, the differences are relatively small, and so the role of context in this model remains to be understood.

## 4.7 Comparison with linear classifier accuracy

Figure SM8: Theoretical capacity (left) matches empirical capacity (center), and trends are also reflected in the generalization error of a linear classifier (right) in DS2. The top row of plots shows results on the LibriSpeech word manifolds dataset, while the bottom shows results on the LibriSpeech speaker manifolds dataset.

As further verification of the MFTMA technique, we also compared the trends in capacity, both theoretical $\alpha_{MFT}$ and empirical $\alpha_{SIM}$ to trends observed in the generalization performance of a linear classifier. Figure SM8 show this comparison to the generalization accuracy of a softmax classifier trained on 80% of the manifold data and tested on the remaining 20%. Here, the classifier was trained on 10 random train/tests splits of the manifold data, and the average over manifolds and trials is reported here along with the standard deviation of the mean. The trends observed in the the classifier performance follow those seen in the measures of capacity.

## 4.8 Control experiments with permuted labels

As a check that the effects we observe are indeed due to the structure of the data within the category label used to define the manifold, we conducted control experiments in which the manifold data remained unchanged, but the catergory labels were permuted. This destroys the manifold structure of data, and so the MFTMA method should give manifold capacity values near the lower bound. Figures SM9 through SM18 demonstrate that this is indeed the case.

Figure SM9: Permuted vs. true manifold labels, CNN word manifolds on word trained network

Figure SM10: Permuted vs. true manifold labels, CNN word manifolds on speaker trained network

Figure SM11: Permuted vs. true manifold labels, CNN word manifolds on multitask trained network

Figure SM12: Permuted vs. true manifold labels, Librispeech words experiment, DS2

Figure SM13: Permuted vs. true manifold labels, Librispeech speakers experiment, DS2

Figure SM14: Permuted vs. true manifold labels, TIMIT words experiment, DS2

Figure SM15: Permuted vs. true manifold labels, TIMIT speakers experiment, DS2

Figure SM16: Permuted vs. true manifold labels, TIMIT characters experiment, DS2

Figure SM17: Permuted vs. true manifold labels, TIMIT phonemes experiment, DS2

Figure SM18: Permuted vs. true manifold labels, TIMIT parts of speech experiment, DS2

## Footnotes

[1]The tool used was the Montreal Forced Aligner https://montrealcorpustools.github.io/Montreal-Forced-Aligner/