[Reviews · NeurIPS 2019]

Reviewer 1



This paper presents an interesting direction towards deeper understanding of speech signals in neural architectures. The methodology adapts an existing approach based on mean field theory to audio and speech recognition tasks Comparing two random weights might be an overly weak baseline. It would be nice to see a comparison to weights after a short time training to skip past the poorly scaled norms of initial weights that can sometimes occur It would be nice if the main figures in the paper could include error bars or auxiliary experiments that show whether the findings presented are robust across different training runs of the same neural architecture I am not sure that other practitioners would be able to implement the metrics used and reproduce the experimental setup with other neural architectures. Even though some information is given in the supplemental material more specific method description would help others continue to use these techniques

Reviewer 2



The paper is overall well written, and the experimental design is fundamentally well thought out and rasonable - I cannot say if it is entirely novel or if similar looking graphs could have been achieved with different or similar techniques. The results look intuitively correct and confirm ones expectations. I find some parts of the experimental setup confusing: the CNN *model* has been trained on WSJ and Spoken Wikipedia, and is not performing an ASR task, but a closed-set word recognition task (in addition to the speaker ID task). Why was Spoken Wikipedia used in addition to WSJ? Would it not have been possible to use Librispeech or another well-known corpus? The entire paper would be much "cleaner" if both types of systems ("CNN"=word recognition + "DS2"=ASR) would have been trained and evaluated on the same type of data. If that is not possible - please explain. What is the "CNN dataset"? It is interesting that training a system towards phones will also increase the capacity for words. Would it be possible to perform the same experiments with characters (which is what the DS2 system has been trained with)? This should exhibit a similar pattern, but one could then also compare phone and character systems. In English, the grapheme to phoneme relationship (phones to characters) is pretty complicated ("tangled"), and it should be possible to show that this analysis can "measure" the degree to which certain phones have clear relationships with characters, and other phones have no unique relationship with characters.

Reviewer 3



From the high-level of review, there are many works on understanding how neural network models do speech recognition internally, and this paper provides very solid analysis using a theoretical tool such as the mean-field theory. However, I'm not fully convinced how much the findings here can benefit the speech recognition research. In particular, the observations are from a particular model configuration (network structure, word-level label units) etc. It is unclear to me how much the observations would change in a different experimental setting, for example, with a system modeling the sub-phonemes as the traditional hybrid system, and evaluating on more challenging conversational speech corpus. The high-level conclusions such as normalization of speaker factors and untangling words in a speech recognition system are not surprising to me as a speech recognition researcher. It is not new, but this paper provides some more theoretical evidence to confirm what we have believed already. A question: I'm a bit confused what is N the ambient dimension in section 2.1? And I'm fully understand why \alpha in section 2.1 is important. If P is the vocabulary size, we will use a softmax layer with size P for classification. What is this to do with \alpha and separating hyperplane?

[Author Response · NeurIPS 2019]

We are grateful to the reviewers for their insightful and constructive comments. We appreciate that each reviewer
found our approach to be a useful step towards a better understanding of ASR systems. Below we respond to the main
questions and concerns, paraphrased for brevity.

**To All Reviewers:** To improve accessibility of the method, we will open source the analysis code, and clarify our
procedures in the revision.

**Reviewer 1 (R1)**: *Random weight baseline*: We highlight a few pieces of evidence that suggest the differences in
geometry are due to learning rather than the changing weight norms: the effects of training vanish in all cases when the
manifold labels are permuted (SM Fig. 14-20), opposing trends are seen in when measuring word or speaker manifolds
(Fig. 2 and Fig. 4), and these trends change based on the task the model is trained on (Fig. 4 top).

*Error bars*: In the present article, we used a single typical training run of two popular open-sourced models (word-CNN
and DS2). This is consistent with the methods of Refs. [6, 7, 25]. We demonstrate the robustness across different
random projections of network features and initializations for the mean-field metric calculations. An example of 95%
confidence intervals calculated in this way is included for word-CNN (Fig A1(a)) and we will update the remaining
figures in the document similarly.

**Reviewer 2 (R2)**: *Training dataset differences for the CNN and the DS2 models*: The discrepancy in datasets is due
to the necessity of word-aligned speech for training the word-CNN, which is not publicly available for LibriSpeech.
The "CNN dataset" is adapted from that used in Ref. [14], which we supplement with words from Spoken Wikipedia
Corpus (SWC) to diversify the word instances and provide more balanced speaker classes for the speaker trained model.
The details of this training dataset construction are explained in SM section 3, and in the revision we will improve the
clarity in the main text when referencing these datasets and models.

*Connection between phoneme and character manifolds*: We agree that our method could be useful to illuminate the
complex relation between characters and phones in ASR systems. Here we share a preliminary result in Fig A1(b-d),
showing that character-level manifolds also emerge across the layers of DS2. The relative increase in the manifold
capacity in the last recurrent layer of DS2 compared to the input layer is slightly larger in characters than phonemes,
consistent with recent findings by Belinkov, Ahmed, and Glass, arXiv:1907.04224 (2019). The further exploration into
what this method can tell us about the grapheme to phoneme relationship is a promising direction for future work but
would be better served as a separate paper.

**Reviewer 3 (R3)**: *Clarification about the scope of our experiments and findings*: While it is true that some of our
results are consistent with current beliefs, our work provides empirical evidence for these beliefs, and goes deeper by
connecting object information to underlying feature geometry as enabled by the theory. The presented results also
include two different ASR models with different motifs (convolutional and recurrent), one trained with word label units
and the other with character sequences, and we see similar behavior in both experimental settings.

*Clarifying ambient dimension and capacity's relation to vocabulary size*: Ambient dimension $N$ is the feature dimension.
If the manifold capacity ($P/N_c$) is large given fixed $P$ (size of vocabulary), then the required feature dimension for
separability, $N_c$, is small, and the classes are linearly separable (they are untangled), as long as $N > N_c$. More insights
on manifold capacity can be found in Ref [24, 25].

*Correspondence of capacity and Word Error Rate (WER)*: R3 raises an interesting question of the direct relationship
between the capacity measures and WER. To address this, we share an additional result on the relationship between
capacity and WER in DS2 over different epochs of the training (Fig A1(e)) showing that as capacity increases, WER
decreases. We also note that the network-level correspondence between the manifold capacity and the accuracy is
shown in the prior work in vision (Ref. [25]).

Figure A1: (a) word manifold capacity in word-CNN, (blue) after training, (black) before training, (b-d) character
manifolds compared with phonemes, words, POS manifolds, (e) WER vs word capacity on the test set

[Meta-Review · NeurIPS 2019]

The authors propose to borrow some recently developed statistical mechanical theory, and apply it to neural networks in the context of speech recognition, to study hidden representations. As noted by the reviewers, findings are already known - the novelty lies more into the application of the theory to ASR. The idea could be possibly extended to other applications in future work.